# A bioinspired surface tension-driven route toward programmed cellular ceramics

Ying Hong [1,5], Shiyuan Liu[1,5], Xiaodan Yang [1,2,5], Wang Hong[3], Yao Shan[1,2], Biao Wang[4], Zhuomin Zhang [1,2], Xiaodong Yan[1,2], Weikang Lin [1,2], Xuemu Li[1], Zehua Peng[1,2], Xiaote Xu [1,2] & Zhengbao Yang [1] ✉

The intriguing biomineralization process in nature endows the mineralized biological materials with intricate microarchitected structures in a facile and orderly way, which provides an inspiration for processing ceramics. Here, we propose a simple and efficient manufacturing process to fabricate cellular ceramics in programmed cell-based 3D configurations, inspired by the biomineralization process of the diatom frustule. Our approach separates the ingredient synthesis from architecture building, enabling the programmable manufacturing of cellular ceramics with various cell sizes, geometries, densities, metastructures, and constituent elements. Our approach exploits surface tension to capture precursor solutions in the architected cellular lattices, allowing us to control the liquid geometry and manufacture cellular ceramics with high precision. We investigate the geometry parameters for the architected lattices assembled by unit cells and unit columns, both theoretically and experimentally, to guide the 3D fluid interface creation in arranged configurations. We manufacture a series of globally cellular and locally compact piezoceramics, obtaining an enhanced piezoelectric constant and a designed piezoelectric anisotropy. This bioinspired, surface tension-assisted approach has the potential to revolutionize the design and processing of multifarious ceramic materials for structural and functional applications in energy, electronics and biomedicine.

Nature serves as an inspiration in designing various engineering and functional structures, particularly at the micro-nanoscale, which often exhibit remarkable properties that surpass those of human-made counterparts[1–3]. Biomineralization in the natural world is an intriguing process in which the mineralized biological materials are usually synthesized under ambient conditions and endowed with intricate hierarchical crystallographic structures[4,5]. To date, a wide range of natural materials with intrinsic structures have been utilized to manufacture structured ceramics, such as sponge[6], wood[7], teeth[8], bone[4,9], and shell[10]. Single-celled diatoms, renowned for their silica frustule, have

been studied due to their diverse multi-scale three-dimensional (3D) structures, large surface area, excellent mechanical strength, and unique optical properties[11]. What is the most fascinating in diatom is the delicate biomineralization of the diatom frustule.

Benefiting from the genetically programmed biomineralization in the construction of the precise structure of frustules, the diatom possesses a variety of frustule morphology, shape, geometry, pore distribution and assembly[12] (Fig. 1a). Taking *Melosira* (a genus of diatoms) for example, the programmed aligned cell assembly provides *Melosira* with unique cell-based chain structures. During the

[1]Department of Mechanical and Aerospace Engineering, Hong Kong University of Science and Technology, Clear Water Bay, Hong Kong, China. [2]Department of Mechanical Engineering, City University of Hong Kong, Hong Kong, China. [3]Institute of Advanced Structure Technology, Beijing Institute of Technology, Beijing, China. [4]Institute of Artificial Intelligence, School of Future Technology, Shanghai University, Shanghai, China. [5]These authors contributed equally: Ying Hong, Shiyuan Liu, Xiaodan Yang. ✉e-mail: zbyang@ust.hk

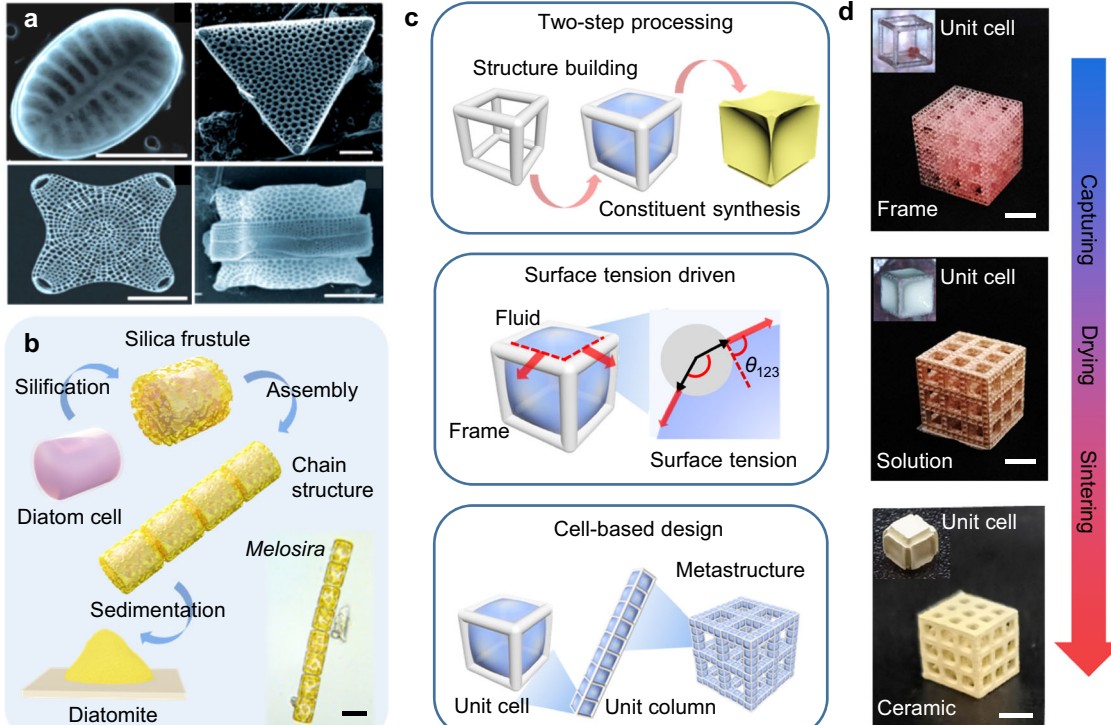

**Fig. 1 | Nature-inspired design of the surface-tension-assisted two-step (STATS) manufacturing process. a** Scanning electron microscopic (SEM) images of diatoms with diverse frustule morphologies. Scale bar, 10 μm. Reproduced with permission from ref. 12, copyright 2009, WILEY-VCH Verlag GmbH & Co. KGaA, Weinheim. **b** The biomineralization of silica frustule and diatomite formation, including the silification process, the assembly of the silica frustule and the sedimentation process. Inset shows the optical image of *Melosira*, a genus of diatoms with chain structure. Scale bar, 20 μm. Inset optical image is modified from http://flickr.com/, licensed under a Creative Common Attribution 2.0 Generic License (https://creativecommons.org/licenses/by/2.0/). **c** The key points of the STATS manufacturing process inspired by the biomineralization of silica frustule, including (i) two-step processing, which separates constituent synthesis from structure building, (ii) surface tension driven, which contributes to the solution capture in programmed arrangement, and (iii) cell-based design, in which the concept of modular design and hierarchical structures is introduced to the microarchitected lattices composed of diverse unit columns, which are further constituted by basic unit cells. **d** The optical images of the architected cellular ceramics from resin frame, captured solution, to sintered ceramics during the manufacturing. Scale bar, 5 mm. Insets show the evolution of the unit cell with an edge length of 1.5 mm.

biomineralization of silica frustule, silicon (primarily in the form of silicic acid, $Si(OH)_4$) is initially actively taken from seawater into the cell by the silicon transporters (SITs). Upon uptake, silicon is transported to the silica deposition vesicle (SDV), where the silification process occurs. Subsequently, the precipitated and polymerized silica is transported through the plasma membrane to form silicified frustule[13,14] (Supplementary Fig. 1). After sedimentation over the years, the widely distributed diatomite, accumulated by the silica-rich diatom fossil skeletons, comes into being[11] (Fig. 1b). Unlike the direct layer-by-layer additive manufacturing of 3D printing ceramics, the biomineralization of the diatom frustule composed of structure building and component synthesis possesses a more orderly and systematic building process. The diatom cell serves as the basic organic skeleton for the structure construction, while the continuous silification process synthesizes silica constituents. Moreover, the programmed assembly of multiple cells endows the diatoms with diverse cell-based 3D structures[12].

High-performance structural and functional ceramics, especially those with interconnected 3D frameworks, have gained increasing attention because of their attractive mechanical and physical properties, such as adjustable strength and toughness, increasing surface area, and excellent electrical and thermal performance[15–18]. These extraordinary properties enable structured cellular ceramics to find a broad range of promising applications for construction, catalysis, biomedicine, sensing, energy harvesting, etc[16]. Recently, the rapid development of 3D printing techniques, e.g., direct ink writing (DIW), stereolithography (SLA), digital light processing (DLP) and selective laser sintering (SLS), provides a promising way for forming programmable geometrically complex and accurate 3D architected ceramics that are either difficult or impossible to realize using conventional approaches such as casting and machining[15,19–23]. However, compared with traditional 3D printing of polymer resin, the additive manufacturing of structured ceramics brings a variety of new problems, such as complex procedure of feedstock preparation, limited ceramic constituent loading, reduced printing accuracy and speed, and increased processing cost.

Inspired by the biomineralization process of the diatom frustule, we design a two-step processing strategy to manufacture cellular ceramics with programmed cell-based 3D architectures, consisting of preparing the cell-based organic lattices assisted by additive manufacturing method to build the basic configurations and filling the precursor solution with required constituents into the architected lattice. The key point lies in how to capture the precursor solution and consequently control the liquid geometry. Surface tension, widely existing in nature, can trap and pin the fluid in prepared lattices[24]. Recently, surface tension-driven fluid interfaces have been utilized in multiple engineering fields related to multiphase processes, such as cellular fluidics[1], nanomembrane processing[25], additive manufacturing[26–30], and object manipulation[24,31–33]. Inspired by these pioneering works, we here introduce the surface tension effect to the area of ceramic processing.

We present a surface tension-assisted two-step (STATS) manufacturing process to efficiently fabricate cellular ceramics in programmed 3D cell-based configurations. The STATS manufacturing

process can prepare globally porous and locally compact cellular ceramics with various cell sizes, geometries, relative densities, 3D metastructures, and constituent elements. We theoretically and experimentally investigate the design principle for the architected lattices composed of unit cells to guide the 3D fluid interface creation and the liquid geometry control. Finally, we demonstrate the practicability of the developed STATS process in piezoceramic manufacturing by systematically characterizing the piezoelectric properties of the prepared cellular piezoceramics. This strategy overcomes the limitations of conventional machining and printing fabrications and provides high freedom for programming complex cellular ceramic architectures.

## Results

### Nature-inspired design by controlling liquid geometry

Inspired by the biomineralization process of the diatom frustule and the formation of diatomite, we design a programmable STATS manufacturing process to efficiently fabricate 3D-architected cellular ceramics. As shown in Fig. 1c, the key points of this approach lie in the following three aspects. (i) We separate constituent synthesis from structure building and design a two-step processing strategy to manufacture cellular ceramics, in which the organic lattices build the basic configurations, and the captured precursor solution consists of the required constituents. (ii) Surface tension drives the solution capture in programmed arrangement and achieves liquid geometry control. (iii) We introduce the concept of modular design and hierarchical structures to the microarchitected lattices composed of diverse unit columns, which are further constituted by basic unit cells. Specifically, the cell-based microarchitected lattice, similar to the diatom cells, is first achieved by additive manufacturing methods. And then, the prepared precursor solution is captured in the lattice by surface tension, which mimics the aggregation of the silicic acid in the diatom cell through an active uptake process. After drying and sintering, corresponding to the silification and sedimentation procedure in the formation of diatom frustule, the cellular ceramics with designed architectures and constituents are finally obtained (Fig. 1d).

For architected ceramics, programmable structure construction and precise ingredient regulation are vital to achieving excellent mechanical and functional properties. In conventional direct 3D printing process, starting with ceramic powder (powder sintering-based SLS) or its composite with additive polymer (slurry extrusion-based DIW or photosensitive resin-based SLA/DLP), structure building and component assembly are conducted in one step, which inevitably causes problems, such as the complex feedstock preparation (e.g., complicated surface modification process to achieve a uniform distribution when mixing ceramic powder with resin in DIW and SLA/DLP), the limited ceramic constituent loading (usually less than 50 vol% for SLA/DLP), the reduced printing accuracy and speed (e.g., twice the exposure time per layer when printing ceramic-resin composites compared with common resin in DLP), and the increasing processing cost (excessive energy consumption in SLS). Supplementary Table 1 summarizes the parameters in terms of effective constituent loading, fabrication speed, and accessible feature size for the well-established DIW, SLA/DLP, and SLS[34–45]. The slurry extrusion-based DIW has the advantage of high effective constituent loading regardless of its limited fabrication speed and feature size, while the photosensitive resin-based SLA/DLP possesses an increasing fabrication speed and decreasing feature size, with limited effective constituent loading.

In comparison, the philosophy of the STATS process is different. It avoids an engineering trade-off by separating the steps of architecture building and ingredient synthesis. The proposed STATS approach combines the advantages of DIW in effective constituent loading and SLA/DLP in fabrication speed and accessible feature size. We utilize SLA/DLP to process high-resolution organic lattices, and we prepare the feedstock suspensions, similar to the DIW process, with high effective ceramic constituent loading. Moreover, the exposure time of common resin is usually 2–3 s per layer in DLP, while it increases over one time (5–7 s) for the curing of DLP processed ceramic-resin mixture[43]. In the proposed STATS approach, the organic lattices with designed architectures are processed by the additive manufacturing of common resin, greatly increasing the printing speed. Therefore, the separation of ingredient synthesis from architecture building in the STATS manufacturing process dramatically simplifies and accelerates the cellular ceramic manufacturing.

### Manufacturing of the cellular ceramic metamaterials

The STATS manufacturing process is shown in Fig. 2a (detailed in Supplementary Fig. 2 and "Methods" section). Here, we choose lead zirconate titanate (PZT), a commonly used functional ceramic, as the synthetic target. To prepare the organic framework used to build the primary structure, we design a series of microarchitected lattices and fabricate them by additive manufacturing methods. The fabricated organic lattices are then immersed in a prepared precursor solution. Once we remove the lattices from the solution to air, part of the precursor solution is captured in the lattices with a programmed arrangement by the surface tension (Supplementary Movie 1, detailed capturing principle will be discussed in the next section). After drying, a sol-gel transition occurs, leading to the shrinkage of precursor solution with a density increase from 2.3 g cm$^{-3}$ to 5.0 g cm$^{-3}$ (Fig. 2b). The obtained gel precursor forms a cellular shell structure, packaging the organic lattice. The final sintering process at high temperature transforms the gel precursor into compact ceramics with a density of 7.0 g cm$^{-3}$, comparable to the commercial PZT ceramic.

To verify that the designed architecture can be maintained during the sintering process, we in situ record the configuration evolution of the prepared samples during the sintering process (Fig. 2c). The results show that the sample architecture remains similar to the initial state during the whole sintering process. An obvious volume shrinkage occurs during sintering, due to the removal of organic ingredients and grain growth of the ceramics. Further differential scanning calorimetry (DSC) and thermogravimetric (TG) analysis reveal that the oxidation reaction of organic components in framework lattices and gel precursor takes place at ~400 °C and ~250 °C, respectively, and the whole weight of the removed organic component in gel precursor occupies less than 10% of the sintered ceramics (Fig. 2c and Supplementary Fig. 3). Compared with the conventional ceramic 3D printing methods (e.g., SLA/DLP), the significantly increased ceramic loading in the STATS strategy can greatly improve the compactness of the sintered ceramics.

The globally cellular shell structure and locally compact ceramic are visualized by the optical and scanning electron microscopic (SEM) images in Fig. 2d and Supplementary Fig. 4. As a comparison, the sample prepared from conventional powder-binder-water based suspensions shows a lower quality with more defects (detailed in Supplementary Fig. 5 and "Methods" section). The mass ratio of the powder to sol solution in the preparation of precursor suspension influences the morphology and compactness of the sintered ceramics, and we choose 2:1 as the optimal mass ratio in the following experiments to achieve a well-densified architected ceramic without significant micropores (Supplementary Fig. 6). During the sol-gel transition, a shrinkage of the precursor solution occurs, leading to a slightly concave surface of the ceramic shell, as shown in the SEM images (Fig. 2d). Moreover, the layer-by-layer characteristic of the ceramic shell reproduced from the 3D-printed organic lattice is exhibited in the SEM section images (Fig. 2e and Supplementary Fig. 5d). The shell thickness is significantly influenced by the concentration of the precursor solution, which increases from 100 μm to 187 μm as the concentration increases from 1 mM to 5 mM (Fig. 2e and Supplementary Fig. 7). Besides the concentration of the precursor solution, the shell thickness and chamber length of the sintered cellular ceramics also depend on the feature size of organic lattice (i.e.,

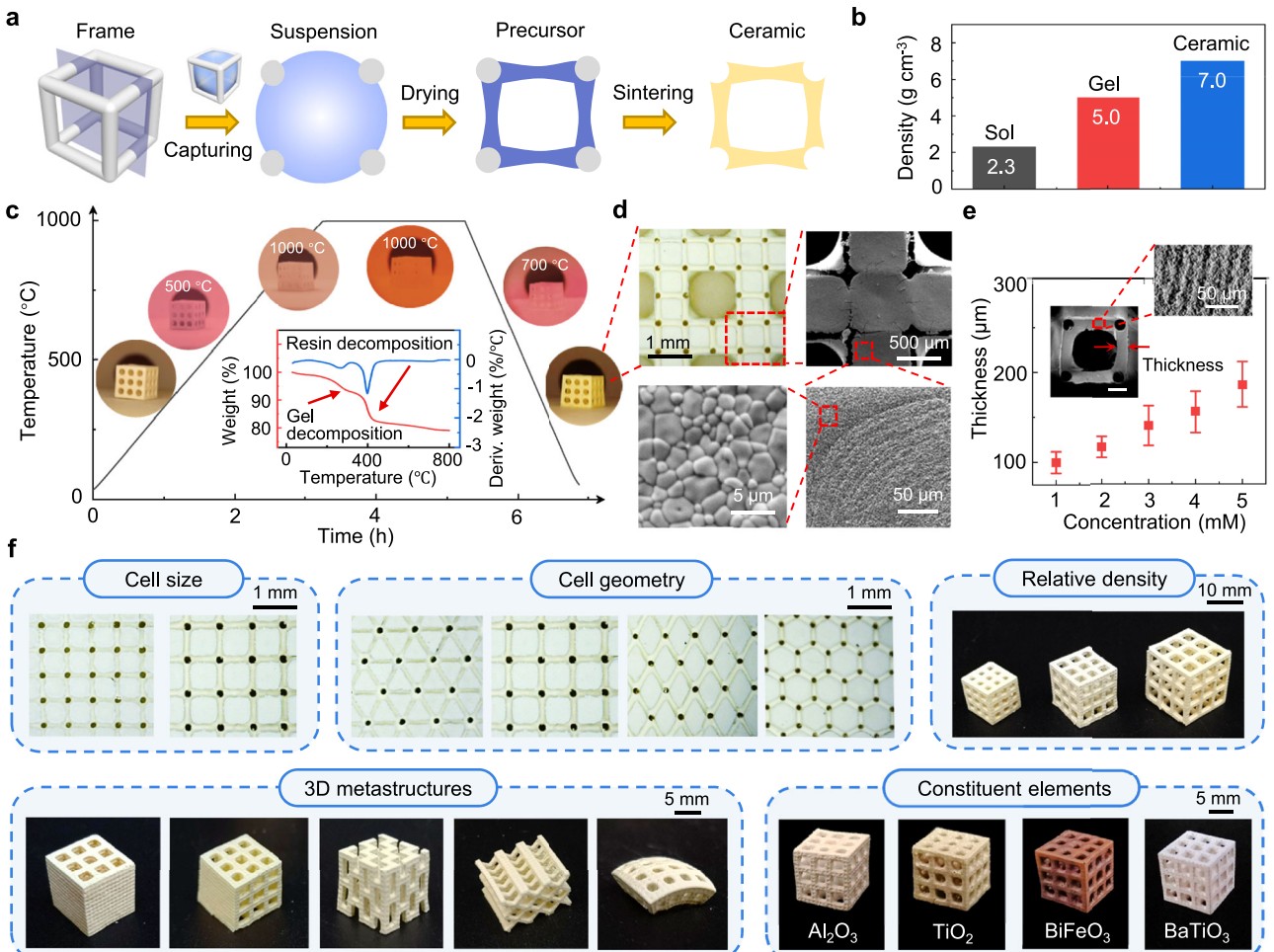

**Fig. 2 | Manufacturing and characterization of cellular ceramic metamaterials. a** Illustration of the STATS manufacturing process in a section view, including precursor solution capture, sol-gel transition, and high-temperature sintering. **b** Density variation during the sol-gel transition and sintering. **c** Temperature profile and thermogravimetric (TG) curves during the sintering process. Inset optical images show the morphology evolution of the samples. **d** Optical images and SEM images of the sintered architected ceramics, showing a globally cellular shell structure and locally compact ceramic grain. **e** The variation of shell thickness

of the sintered cellular ceramics along with the precursor solution concentration, at a 1 mm frame length and 0.1 mm frame radius of the organic lattice. Error bars are standard deviation and include 16 independent measurements for each composition. Insets show the SEM images of the cellular ceramics with shell structure and layer-by-layer characteristic. Scale bar, 200 μm. **f**, The manufactured cellular ceramics with various cell sizes, cell geometries, relative densities, three-dimensional (3D) metastructures and constituent elements.

frame length and radius). Once the frame radius of the organic lattice is reduced from 100 μm to 75 μm, a minimum shell thickness of 65 μm is achieved with a length/thickness ratio of 10 (Supplementary Fig. 8).

To further characterize the cellular structure of the STATS manufactured ceramics, X-ray computed microtomography (micro-CT) is conducted. As shown in Supplementary Fig. 9, the micro-CT images clearly show the globally cellular shell structure of the sintered ceramics with a high quality. There are mainly two types of defects, including the cracks existing in the interface between the ceramic shell and the removed organic lattices, and the extra ceramic walls between the unit cells. These defects have little influence on the overall performance of the sintered cellular ceramics. The energy dispersive spectroscopic (EDS) mapping (Supplementary Fig. 10) shows that the elements of Pb, Zr, and Ti have a homogeneous distribution in the architected ceramics. The perovskite structure of manufactured PZT ceramics is verified by X-ray diffraction (XRD) patterns (Supplementary Fig. 11a) and Raman spectrum (Supplementary Fig. 12b) (detailed characterization process can be found in "Methods" section). As shown in Fig. 2f and Supplementary Fig. 12, the cellular ceramics with various cell sizes, cell geometries, relative densities, 3D metastructures, and constituent elements are manufactured, fully proving that the developed STATS method

possesses high programmability and feasibility in preparing both of the structural ceramics (e.g., $Al_2O_3$) and functional ceramics (e.g., $TiO_2$, $BiFeO_3$, $BaTiO_3$) (detailed manufacturing process of these ceramics can be found in "Methods" section).

## Design principle of the architected lattice

In order to ensure that the precursor solution is captured in the architected lattice with a programmed arrangement, we have to figure out the design principle of the architected lattice (detailed numerical calculation can be found in Supplementary Text 1). We first consider the case of a unit cell. As the unit cell moves from the prepared solution (fluid 1) to air (fluid 2), the solution is captured in the solid frame (solid 3) only when $E_1 < E_2$, where $E_1$ and $E_2$ are the interfacial free energy for solution capture and solution collapse, respectively[24]. This equation can be converted into

$$S_{12} - S_{13}\cos\theta_{123} < 0 \qquad (1)$$

where $\theta_{123}$ is the contact angle of the interface among fluid 1, fluid 2, and solid 3, $S_{12}$ is the interface area of fluid 1 and fluid 2, and $S_{13}$ is the interface area of fluid 1 and solid 3 (Fig. 3a).

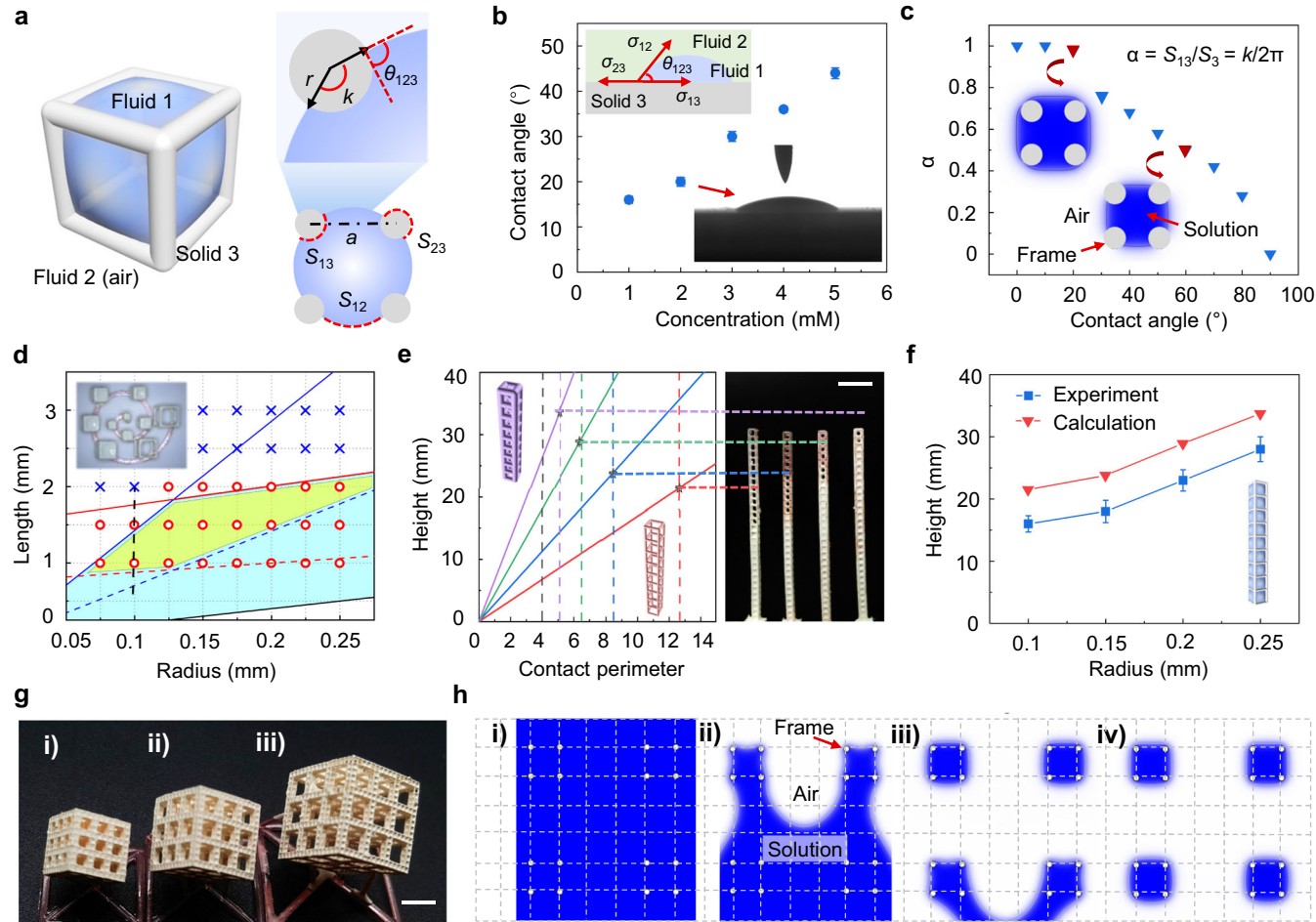

**Fig. 3 | Design principle of the architected lattice. a** Schematic diagram of the liquid capture. Precursor solution (fluid 1) is captured inside a solid frame (solid 3) in air (fluid 2), where $a$ is the frame length, $r$ is the frame radius, $k$ is the central angle corresponding to the interface arc of fluid 1 and solid 3, $\theta_{123}$ is the contact angle of the interface among fluid 1, fluid 2, and solid 3, and $S_{12}$, $S_{13}$, and $S_{23}$ are the interface area of fluid 1 and fluid 2, fluid 1 and solid 3, and fluid 2 and solid 3, respectively. **b** The variation of contact angle along with the concentration. Error bars are standard deviation and include 6 independent measurements for each composition. Insets show the illustration and optical image of the contact angle among fluid 1, fluid 2, and solid 3, where $\sigma_{23}$, $\sigma_{12}$, $\sigma_{13}$ are the interfacial tension of fluid 2-solid 3, fluid 1-fluid 2 and fluid 1-solid 3 respectively. **c** FE results of the dependence of α on the contact angle $\theta_{123}$ for precursor solution in air, where α is the ratio of $S_{13}$ to $S_3$ and $S_3$ is the surface area of solid frame. Insets show the liquid distribution at the contact angle of 20° and 60°. **d** Theoretical and experimental results showing the influence of frame length $a$ and radius $r$ on the liquid capture. The blue and yellow regions respectively include the

appropriate geometry parameters for successful liquid capture in unit cell and liquid arrangement in architected lattice from theoretical analysis. The red circle and blue cross icons indicate successful and failed liquid capture in the experiments, respectively. Inset shows the optical image of the experimental results (black dash line) with various frame lengths from 0.4 mm to 2.2 mm and the same radius of 0.1 mm. **e** The equilibrium point (star icons) shifts with the variation of frame radius (0.1 mm, 0.15 mm, 0.2 mm, 0.25 mm). Inset shows the optical image of the experimental results. Scale bar, 5 mm. **f** Calculated results of the liquid height dependent on the frame radius, agreeing well with the experimental results. Error bars are standard deviation and include 5 independent measurements for each radius. **g** Optical image of the architected lattices with precursor solution filled in a programmed arrangement. The length of the unit cell is $a$, and the interval lengths are (i) $2a$, (ii) $3a$, and (iii) $4a$, respectively. Scale bar, 5 mm. **h** Finite element analysis of the formation of the programmed liquid arrangement from (i) to (iv) under gravity for the architected lattice with an interval length of $3a$.

Equation 1 indicates $\theta_{123} < 90°$, so the cube frame should be lyophilic in air. We characterize the contact angle of the precursor solution on the resin plate in air. The contact angle increases, accompanied by the increasing concentration of the precursor solution, but all within 90° (Fig. 3b and Supplementary Fig. 13). Taking into account the contact angle and the volume shrinkage in drying and sintering, we choose the precursor solution with a concentration of 2 mM and a contact angle of 20° in the following experiments. Besides, the addition of surfactants in the precursor solution and the plasma treatment on the organic lattices also benefit the decrease of contact angle to some extent (Supplementary Fig. 14 and Supplementary Text 2). Considering that a contact angle of 20° is enough to form a lyophilic condition, these factors are not considered in the following

experiments from the perspective of simplification of experimental procedures.

Meanwhile, $S_{12} < S_{13}$ can be derived as $\cos\theta_{123} \leq 1$ and a larger $S_{13}$ benefits the liquid capture. We define α as the ratio of $S_{13}$ to $S_3$, where $S_3$ is the surface area of solid frame. To study the influence of α, we utilize finite element (FE) analysis to quantify its value (Supplementary Fig. 15 and Supplementary Movie 2, detailed in "Methods" section). As shown in Fig. 3c, α decreases from 1 to 0 when the contact angle increases from 0° to 90°. At the contact angle of 20°, α is 0.98, close to its maximum, which helps to form the liquid interface.

To better elaborate the detailed design principle, we take a cube cell with a frame length of $a$ and a frame radius of $r$ for example

(Fig. 3a). Combining Eq. 1 and the geometry requirement ($a > 2r$), we obtain $2.0 < a/r < 14.3$, shown as the blue solid line and black solid line in Fig. 3d. Besides, the size of the created fluid interface ($a - 2r$) must be within the capillary length $\sqrt{\sigma_{12}/\triangle\rho_{12}g}$, where $\sigma_{12}$ is the interfacial tension of fluid 1-fluid 2, $\Delta\rho_{12}$ is the density difference between fluid 1 and fluid 2, and $g$ is the gravitational acceleration[24]. The obtained result ($a - 2r < 1.64$ mm) is shown as the red solid line in Fig. 3d, and the blue region represents the safe area with the appropriate frame length and radius to successfully create a stable liquid interface. Further experiments also agree with the theoretical analysis (Fig. 3d and Supplementary Fig. 16).

Besides the frame length and radius of the unit cell, the height of the unit column is another parameter to be considered under the influence of gravity. According to Jurin's law[1], the liquid height in the unit column can be estimated by the equilibrium of surface tension force and gravitational force, which is given as

$$h = \frac{\sigma_{12}cos\theta_{123}s}{\rho g a^2(1 - \varphi)} \qquad (2)$$

where $h$ is the liquid height, $s$ is the liquid-solid contact perimeter, $\rho$ is the liquid density, and $\varphi$ is the volume ratio of the solid frame to the unit cell. The liquid-solid contact perimeter varies with the frame radius and the top plane position of the liquid, and always reaches a maximum when contacting with the top frame of the cell (Supplementary Figs. 17a, b). According to Eq. 2, under the equilibrium state, a larger contact perimeter $s$ results in a larger liquid height $h$. Figure 3e summarizes the relation between liquid height and the maximum contact perimeter in a unit column as the radius increases from 0.1 mm (red dash line) to 0.25 mm (purple dash line) (detailed in Supplementary Fig. 17c). A relation between the liquid height $h$ and the contact perimeter $s$ with corresponding radii from the gravitational force is also given as the solid line in Fig. 3e (e.g., red solid line corresponds to a radius of 0.1 mm) according to Eq. 2. The increasing frame radius leads to a higher maximum contact perimeter and volume ratio $\varphi$, corresponding to a higher surface tension force and a lower liquid gravitational force. As a result, the equilibrium point (star icons in Fig. 3e) shifts with increasing frame radius, contributing to a higher liquid height (Fig. 3f). Further experiments also verify the theoretical results (Figs. 3e, f). In a word, a larger frame radius benefits a broader range of liquid height when capturing liquid in a unit column.

When considering the whole architected lattice, in which the precursor solution is assembled with a programmed arrangement, both the successful capture in parts of the unit cells and the failed capture in other parts are required. Taking the architected lattices depicted in Fig. 3g for example, the unit cells with a length of $a$ should be filled with the precursor solution while it is the opposite for the cells with a length of (i) $2a$, (ii) $3a$, and (iii) $4a$. As a result, the constraints of $7.15 < a/r < 14.3$ and $0.82$ mm $< a - 2r < 1.64$ mm (the yellow region in Fig. 3d) are required to form a programmed arrangement with an interval length of (i) $2a$ depicted in Fig. 3g. FE analysis is further utilized to verify the calculated and experimental results, as shown in Fig. 3h, Supplementary Fig. 18 and Supplementary Movie 3.

**Piezoelectric performance of the cellular ceramic metamaterials**
To elaborate on the superiority of the developed STATS manufacturing process, we choose cellular piezoceramics and characterize their piezoelectric performance. The processing difficulty of cellular piezoceramics lies in the dilemma between geometrical complexity and piezoelectricity[46]. The proposed STATS manufacturing approach can prepare cellular ceramics with programmed architectures, decrease the micropores, and improve the local compactness in the sintered cellular ceramics because of the significantly reduced organic component in the feedstock. This process thus benefits manufacturing globally porous and locally compact cellular piezoceramics with anticipated piezoelectric performance.

Using PZT piezoceramics as a proof of concept, we systematically characterize the piezoelectric properties of solid and cellular PZT piezoceramics through STATS manufacturing process. The as-prepared solid piezoceramic plates exhibit a piezoelectric constant ($d_{33}$) around 500 pC N$^{-1}$, obviously higher than that of the previously reported architected piezoelectric composites or ceramics[37,47–54] (Fig. 4a). Such a high $d_{33}$, close to commercial PZT, proves that the developed STATS strategy can be used to manufacture highly compact piezoceramics with excellent piezoelectricity.

To further demonstrate the advantages of the STATS approach in manufacturing cellular piezoceramics, we fabricate a series of cellular PZT ceramics with various relative densities (relative to the density of solid ceramics) and characterize their mechanical and piezoelectric properties. Under uniaxial compression test, the fracture strength of the cellular PZT ceramics increases from 1.0 MPa to 3.9 MPa as the relative density increases from 0.13 to 0.34 (Fig. 4b and Supplementary Fig. 19). The fracture strength of the cellular ceramics with different constituents is further characterized (e.g., 1.6 MPa for BaTiO$_3$ and 5.8 MPa for Al$_2$O$_3$, as shown in Supplementary Fig. 20), comparable to the 3D-printed porous ceramics previously reported[55].

The piezoelectric performance of the prepared cellular PZT ceramics is demonstrated by characterizing their $d_{33}$ and piezo-electric voltage coefficient ($g_{33}$). Owing to the similar configuration, the cellular PZT ceramics exhibit a close $d_{33}$ (~ 200 pC N$^{-1}$) regardless of the changing relative density from 0.13 to 0.34 (Supplementary Fig. 21). Compared with the solid piezoceramic plates, the decreased $d_{33}$ is attributed to the high porosity. Nevertheless, the locally compact structure ensures that the STATS manufactured cellular piezoceramics possess a relatively high $d_{33}$ even at a very high overall porosity (> 90%). On the other hand, the effective $g_{33}$, quantified through dividing $d_{33}$ by permittivities of the as-prepared metamaterials, increases remarkably with reduced relative density because of the decrease of permittivities (Fig. 4b, Supplementary Fig. 21, and Supplementary Text 3). The high effective $g_{33}$ (0.89 V m N$^{-1}$ at a relative density of 0.13) indicates a potential application in highly sensitive piezoelectric voltage-type sensors[50]. We further characterize the piezoelectric responses under periodical compressive pressure, as shown in Fig. 4c. At a compressing frequency of 40 Hz, the cellular piezoceramic with a relative density of 0.13 exhibits a pressure sensitivity of 3.2 mV kPa$^{-1}$. The globally porous and locally compact structure originating from the STATS manufacturing process endows the cellular piezoceramics with a relatively low density and a higher piezoelectric constant than a variety of piezoelectric ceramics/composites[56–59] (Fig. 4d).

Compared with the conventional direct 3D printing process of cellular piezoceramics (e.g., SLA/DLP, DIW, and SLS), the proposed STATS manufacturing process in this work demonstrates multifarious advantages in ceramic compactness, geometry complexity, processing precision, processing speed, total cost, and piezoelectricity (Fig. 4e). To further evaluate the piezoelectric responses of the designed cellular piezoceramic metamaterials, we prepare a series of cellular piezoceramics with different 3D configurations based on the programmed architected lattices composed of periodic unit cells. The structural anisotropy results in the piezoelectric anisotropy[60]. Three types of cellular piezoceramics are selected and polarized along the three principal directions, as shown in Fig. 4f. Under the periodical compressive pressure from different directions, the polarized cellular piezoceramics output corresponding electrical signals. For type i cellular ceramics with the same configuration along the three principal directions, similar output voltage values are obtained under pressure from different directions. However, for type ii and type iii cellular ceramics with varied configurations along the three principal directions, besides mechanical anisotropy

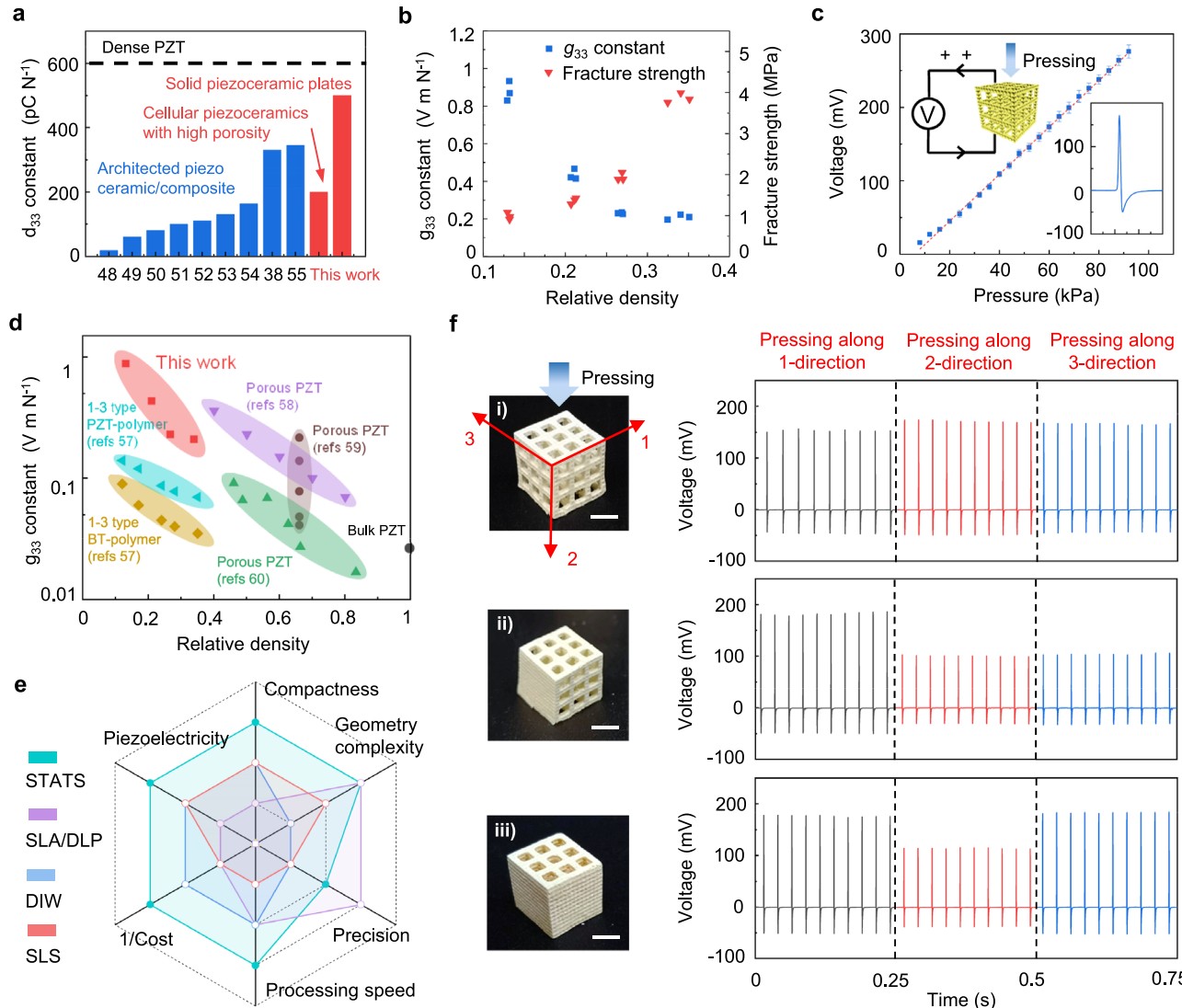

**Fig. 4 | Piezoelectric performance of cellular ceramic metamaterials.**
**a** Comparison of the $d_{33}$ constant between the STATS manufactured piezoceramics including solid piezoceramic plates and cellular piezoceramics with high porosity and the previously reported architected piezoelectric composites and ceramics[37,47–54]. **b** The effective $g_{33}$ constant and fracture strength of the prepared cellular piezoceramics with an average relative density of 0.13, 0.21, 0.27 and 0.34. **c** The output voltages of the cellular piezoceramics with a relative density of 0.13 under increasing periodical compressive pressure. The load frequency is 40 Hz. Error bars are standard deviation and include 6 independent measurements for each pressure. Insets show the current direction during the compression and the typical voltage signals during 7.5 ms under the pressure of 60 kPa. **d** Comparison of the $g_{33}$ constant between the STATS manufactured cellular piezoceramics and the previously reported piezoelectric materials with various relative densities[56–59]. **e** Comparison of the proposed STATS manufacturing process with the conventional direct 3D printing process, i.e., stereolithography (SLA) or digital light processing (DLP), direct ink writing (DIW), and selective laser sintering (SLS), on engineering architected piezoceramics in terms of ceramic compactness, geometry complexity, processing precision, processing speed, total cost, and piezoelectricity. **f** Optical images of the architected piezoceramic metamaterials with different 3D configurations and their corresponding output voltage signals under periodical compressive pressure from the three principal directions. Scale bar, 5 mm.

(Supplementary Fig. 22), there is also an obvious difference in the output voltages under pressure from different directions, demonstrating the piezoelectric anisotropy.

## Discussion

This work presents a facile and efficient strategy to manufacture cellular ceramics in programmed 3D configurations by leveraging the surface tension effect. The main innovation of this approach lies in the following three aspects. (i) Inspired by the biomineralization process of the diatom frustule, we creatively separate ingredient synthesis from architecture building, dramatically simplifying and speeding up the manufacturing process of cellular ceramics. (ii) Based on architected cellular lattices, we introduce the surface tension effect to the process and further control the liquid geometry, achieving

programmable cellular ceramic manufacturing. (iii) We adopt a cell-based modular design concept for the microarchitected lattices, significantly facilitating the design and modeling process.

Although the surface tension effect has been well described and applied in multifarious areas, to the best of our knowledge, no report exploits this physical phenomenon for ceramics manufacturing in this manner. The developed STATS manufacturing strategy overcomes the limitations of conventional manufacturing methods and offers an accessible and versatile platform for efficiently creating programmable and geometrically complex ceramic architectures. We envisage that this approach could help process numerous structural and functional cellular ceramics and contribute to applications involving filters, sensors, actuators, robotics, battery electrodes, solar cells, and bactericidal devices. The philosophy of engineering fluid interface for

solid fabrication also provides a new solution for combining interfacial processing with innovative manufacturing, enlightening the synergistic development of advanced design and intelligent materials.

## Methods

### Processing procedure of STATS manufactured cellular PZT ceramics

**Preparation of the cellular microarchitected lattices.** All cellular microarchitected lattices are printed with a commercial DLP printer (JC2-M10, JANSUM) using the commercial resin JC-H-00. All the lattices are printed with 20-μm layers. After rinsed in isopropyl alcohol, the printed lattices are post-cured through exposed to ultraviolet light.

**Preparation of the PZT [$Pb(Zr_{0.52}Ti_{0.48})O_3$] sol solution.** Lead (II) acetate trihydrate (99%, Dieckmann), zirconium (IV) propoxide (70 wt% in 1-propanol, TCI), titanium (IV) butoxide (99%, TCI) and acetic acid (99%, Dieckmann) are prepared as the starting materials. The lead (II) acetate trihydrate with a 20% excess is firstly distributed in acetic acid under magnetic stirring at 70 °C until fully dissolved. After cooling down to room temperature, the mixed solution of zirconium (IV) propoxide and titanium (IV) butoxide is dropped slowly into the prepared solution under vigorous stirring. The concentration of the final PZT sol is adjusted by controlling the amount of acetic acid, with a molar ratio of Pb: Zr: Ti = 1.2: 0.52: 0.48. A 20% excess of Pb is used for the compensation of lead volatilization in high temperatures. Fully stirring for 0.5 h, deionized water is then added to the prepared solution to stabilize the PZT sol solution.

**Preparation of the PZT sol-powder suspension.** To prepare the PZT sol-powder suspension, PZT powders need to be modified first. The purchased PZT powders (Yisheng Electronics Co., Ltd, Xi'an) and polyethyleneimine (PEI, Aldrich) are distributed in distilled water separately. Then the prepared PEI solution is added slowly to the dispersed PZT-water suspension with vigorous stirring. After fully stirring, the surface-modified PZT powders are successfully separated by centrifugation, and the extra PEI solution is washed away by distilled water. After the hydrophilic treatment mentioned above, the surface-modified PZT powders are dispersed in the prepared PZT sol solution with predefined mass ratio. After stirring for 2 h, the well-dispersed PZT suspension is obtained.

**Preparation of the STATS manufactured cellular PZT ceramics.** The 3D-printed cellular microarchitected lattice is immersed in PZT suspension until the whole lattice is infiltrated with the PZT suspension. The sample is then removed to air and dried in an oven at 70 °C for 1 h. After a sol-gel transition, the gel precursor is sintered in 1000 °C for 2 h at Pb atmosphere to remove the organic lattice and transform the gel precursor into compact ceramics.

### Processing procedure of conventional powder-binder-water based suspensions

To prepare the conventional powder-binder-water based suspensions, the PZT powders are dispersed in deionized water to form a suspension with a solid loading of 40 vol%, in which polyvinyl alcohol (PVA) and polyacrylic acid serve as binder and dispersant respectively. The prepared suspension is ball-milled for 24 h in zirconia media to create a homogeneous suspension, which is used in the following STATS manufacturing process.

### Processing procedure of STATS manufactured $Al_2O_3$, $TiO_2$, $BiFeO_3$, $BaTiO_3$ ceramics

**Preparation of $Al_2O_3$ ceramics.** Tris(2, 4-pentanedinonato)aluminum (III) (99%, TCI) is distributed in acetic acid (99%, Dieckmann) to obtain the $Al_2O_3$ precursor solution. The prepared solution is then mixed with the modified $Al_2O_3$ powders (99.99%, 400 nm, Macklin) to obtain the $Al_2O_3$ suspension. The 3D-printed cellular microarchitected lattice is immersed in $Al_2O_3$ suspension and then removed to air. After drying in an oven at 70 °C for 1 h, the precursor is sintered in 1300 °C for 2 h to transform the precursor into compact ceramics.

**Preparation of $TiO_2$ ceramics.** Titanium (IV) butoxide (99%, TCI) is diluted in acetic acid (99%, Dieckmann) and then mixed with the modified $TiO_2$ powders (100−300 nm, Macklin) to obtain the $TiO_2$ suspension. The 3D-printed cellular microarchitected lattice is immersed in $TiO_2$ suspension and then removed to air. After drying in an oven at 70 °C for 1 h, the precursor is sintered in 1200 °C for 2 h to transform the precursor into compact ceramics.

**Preparation of $BiFeO_3$ ceramics.** The equimolar amounts of bismuth nitrate ($Bi(NO_3)_3.5H_2O$, 99%, TCI) and iron (III) nitrate ($Fe(NO_3)_3.9H_2O$, 99%, TCI) are respectively distributed in 2-methoxyethanol (Alfa Aesar, 99%) and deionized water under magnetic stirring at 70 °C until fully dissolved. An excess of Bi is used for the compensation of Bi volatilization in high temperatures. Acetic acid (99%, Dieckmann) is added in the solution to control precipitation, with water and ethylene glycol (99.5%, Dieckmann) serving as reaction medium and inhibitor, respectively. $BiFeO_3$ powders are synthesized by conventional solid-state process. The raw powders $Bi_2O_3$ (99%, Sinopharm Chemical Reagent Co.) and $Fe_2O_3$ (99.9%, Macklin) are weighed with an equimolar amount. These raw materials are mixed through a planetary ball mill at 400 r min$^{-1}$ for 3 h. The dried mixture is calcined at 800 °C for 4 h to form the perovskite structure. Then the powders are ball milled again for 6 h to refine the particle size. The prepared $BiFeO_3$ sol solution is then mixed with the modified $BiFeO_3$ powders to obtain the $BiFeO_3$ sol-powder suspension. The 3D-printed cellular microarchitected lattice is immersed in $BiFeO_3$ suspension and then removed to air. After drying in an oven at 70 °C for 1 h, the obtained gel precursor is sintered in 1000 °C for 2 h to transform the gel precursor into compact ceramics.

**Preparation of $BaTiO_3$ ceramics.** Barium (II) acetate (99%, TCI) is firstly distributed in acetic acid (99%, Dieckmann) under magnetic stirring at 70 °C until fully dissolved. After cooling down to room temperature, an equimolar amount of titanium (IV) isopropoxide (98%, Alfa Aesar) is dropped slowly into the prepared solution under vigorous stirring. The concentration of the final $BaTiO_3$ sol is adjusted by controlling the amount of acetic acid. Fully stirring for 0.5 h, deionized water and ethylene glycol (99.5%, Dieckmann) are then added to the prepared solution to stabilize the $BaTiO_3$ sol solution. The prepared $BaTiO_3$ sol solution is then mixed with the modified $BaTiO_3$ powders (99.9%, 500 nm, Macklin) to obtain the $BaTiO_3$ sol-powder suspension. The 3D-printed cellular microarchitected lattice is immersed in $BaTiO_3$ suspension and then removed to air. After drying in an oven at 70 °C for 1 h, the obtained gel precursor is sintered in 1200 °C for 2 h to transform the gel precursor into compact ceramics.

### Materials characterization

The scanning electron microscopic morphologies are examined by a scanning electron microscope (SEM, FEI Quanta 450). The energy dispersive spectroscopic (EDS) mapping is characterized by an energy dispersive spectrometer (Oxford Instruments, INCA Energy 200). The crystal structure is characterized by X-ray diffraction (XRD, Rigaku SmartLab). The Raman spectrum is characterized by a Raman spectrometer (Perkinelmer Raman station 400 F). The differential scanning calorimetry (DSC) and thermogravimetric (TG) curves are conducted by a thermogravimetric analyzer (TGA/DSC 3 + , METTLER TOLEDO). X-ray computed microtomography (micro-CT) is conducted by a submicron X-ray microscope (Zeiss XRadia 520 Versa), with a voxel size of 10 μm. The contact angles are measured by Standard Contact Angle Goniometer (OCA25, Dataphysics). The surface tension is

measured by a surface tensiometer (JYW-200A, LARYEE). The compressive stress-strain curves are characterized by a universal platform.

## Finite element simulation of the liquid capturing process

The FEM simulations of the creation of water-in-air interface with the cube cell (Supplementary Movie 2) and the architected lattice (Supplementary Movie 3) are performed using COMSOL Multiphysics 5.5. The two-phase flow and phase-field module are used. The contact angle of solution on the cube frame varies from 0° to 90° and the corresponding $\alpha$ is calculated, as shown in Fig. 3c, Supplementary Fig. 15 and Movie 2. The liquid arrangements for the architected lattice with various interval lengths are further simulated, as shown in Fig. 3h, Supplementary Fig. 18 and Movie 3. The 2D model is used for simplifications.

## Piezoelectric performance characterization

To characterize the piezoelectric performance, the manufactured solid PZT ceramic plates and cellular PZT ceramics are coated by the silver paste on both sides as the electrodes and then polarized in oil bath at 130 °C for 1 h under the electric field of 5 kV/mm.

The quasi-static piezoelectric constants of the prepared solid piezoceramic plates are measured with a quasi-static piezoelectric meter (YE2730A d33 meter). The piezoelectric performance characterization of the manufactured cellular PZT ceramics is conducted by a vibration generator, with a controllable oscillation frequency and compressing force. The compressing force, detected and quantified by a mechanical force sensor, is adjusted by the distance between the taping pillar and sample surface. The piezoelectric output voltage and charge are measured by a digital oscilloscope (Rohde & Schwarz RTE1024) and an electrometer (6517B, Keithley), respectively. Figure 4c shows the generated voltage as a function of impact force and the slope equals the pressure sensitivity.

The $d_{33}$ constants of the cellular PZT ceramics are verified by the ratio of the applied load and the generated charge. And the $g_{33}$ constants are further quantified by the ratio of electric permittivity $\varepsilon_{33}$ and $d_{33}$ constant. The electric permittivity can be calculated from the capacitance of the samples which is measured by a commercial LCR meter (4091 C, VICTOR).

## Data availability

The data generated within the main text are provided in the Source Data. Additional raw data are available from the corresponding authors upon request. Source data are provided with this paper.

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

## Acknowledgements

The work described in this paper was supported by the Hong Kong Research Grants Council (GRF Project No. 11212021 and No. 11210822), and the Innovation and Technology Fund (Project No. ITS/065/20; GHP/096/19SZ) from Innovation and Technology Commission of Hong Kong Special Administrative Region.

## Author contributions

Y.H. and Z.Y. conceived the project and designed the studies. Y.H., S.L. and X.Y. performed experiments and analyzed the experimental data. Y.H., Y.S. and B.W. worked on simulation, with assistance from W.H., Z.Z., X.Y., W.L., X.L., Z.P. and X.X. in materials characterization. Y.H., S.L., X.Y. and Z.Y. composed the manuscript. All authors discussed the results and commented on the manuscript. Y.H., S.L. and X.Y. contribute equally to this study and share the first authorship. All authors approved the submission.

## Competing interests

The authors declare no competing interests.
