## [Peer Review File · Nature Communications]

A Bioinspired Surface Tension-driven Route toward Programmed Cellular CeramicsREVIEWER COMMENTS

Reviewer #1 (Remarks to the Author):

The manuscript presents a novel and intriguing way of additive manufacturing of cellular ceramics. A 3D printed organic cellular structure is used as a scaffold and via a variation of the surface tension of the pre-cursor solution the structure of the final cellular ceramics can be tuned in a certain range, the balance between surface tension and gravity is another physical constraint determining the accessible range of structural specifications. Overall, the processing route is described quite well and as far as I understand the final result is an open cellular ceramic object consisting of densely sintered struts. The paper may be considered for publication in Nature Communications, but requires significant revision.

1) The presented cellular ceramic structures (see Fig. 4 or Suppl. Fig. 8) could also be produced in a simpler and faster way using direct ink writing (DIW). There is a plethora of literature about that, and e.g. the groups of Jennifer Lewis (Harvard University), Andre Studart (ETH Zürich) or Willenbacher (KIT) have published significant work in that title and the authors should benchmark their new approach against well-established DIW in terms of fabrication speed or accessible feature size. Furthermore, the first paragraph of the Results section should be moved to the Introduction.

2) The wording in the section about the design principle on page 5 could be clearer e.g. in line 211 it should read "...increasing precursor solution concentration..." and the term "capillary length" is not clearly defined (line 224).

3) The authors have varied the precursor solution concentration in order to change the wetting angle. This may have an effect on drying time, gel-structure, sintering conditions and also the properties of the final ceramics. Alternatively, the wetting angle could be changed in a wide range by adding trace amounts of appropriate surfactants to a precursor solution with fixed concentration, also the surface properties of the 3D printed organic scaffold may be varied (e.g. plasma treatment) to tune the wetting behavior, these aspects should be discussed carefully.

4) Caption of Fig. 3 is incomplete: The insert in Fig. 3d is not visible (not even when enlarged on the computer screen), the meaning of the different lines in Fig. 3d is unclear. The colors of the lines in Fig. 3e are not explained, correlation to frame radius? Meaning of blue regions in Fig. 3h is not explained, meaning of subfigures i), ii), iii, and iv) in Fig. 3h is not explained.

5) The section about the piezoelectric performance of the cellular ceramic metamaterials is the weakest part of the manuscript. It is not clear what could be achieved here beyond the state of the art regarding piezoelectric properties, e.g. in David Menne et al., ACS Applied Materials and Interfaces 14 (2), 3027-3037 (2022) cellular piezoceramic structures are described with higher d_{33} values and it is demonstrated in Fig. S1 of that paper, that strut diameters as small as 70 μm can be achieved using DIW.

6) Fig. 4 is confusing, in Fig. 4a a d_{33} value of about 500 pC/N is claimed (red bar) but in Fig. 4b d_{33} values are only about 200 pC/N which is clearly lower than the value of 350 pC/N reported by David Menne et al. (see above) On page 7, line 287 a pressure sensitivity of 3.2 mV/kPa is mentioned (slope of Fig 4c) the authors should explain how this quantity is related to the extensively discussed g_{33} data.

Reviewer #2 (Remarks to the Author):

The manuscript discusses a novel approach for manufacturing cellular ceramics inspired by the biomineralization process found in nature. This approach involves separating ingredient synthesis from architecture building to create programmable 3D configurations of cellular ceramics. The

technique utilizes surface tension to capture precursor solutions in architected cellular lattices, allowing for precise control of liquid geometry. The authors investigate the geometry parameters of these lattices theoretically and experimentally to guide 3D fluid interface creation. They successfully manufacture various cellular ceramics, including piezoceramics with enhanced properties, such as a higher piezoelectric constant and designed anisotropy.

The proposed approach possesses similarities to the negative replica approach used for foams as well as for printed sacrificial structures (see for example: <https://doi.org/10.1111/j.1151-2916.1997.tb02993.x>); nevertheless, the theoretical approach proposed here allows to predict and design structures specifically to fit the process requirements and is certainly novel.

I would recommend to further characterize the cellular structures produce to understand how they fit in the current state of the art; in particular:

- microCT scan and defect assessment;
- mechanical properties (not only piezoelectric ones)

I would also better address the role of the usage of sol-gel precursors in achieving the proposed results: what happens if conventional powder-based suspensions are used instead?

RESPONSE TO REVIEWERS' COMMENTS

Reviewer Comments (and change made in accordance)

Reviewer #1 (Remarks to the Author):

The manuscript presents a novel and intriguing way of additive manufacturing of cellular ceramics. A 3D printed organic cellular structure is used as a scaffold and via a variation of the surface tension of the pre-cursor solution the structure of the final cellular ceramics can be tuned in a certain range, the balance between surface tension and gravity is another physical constraint determining the accessible range of structural specifications. Overall, the processing route is described quite well and as far as I understand the final result is an open cellular ceramic object consisting of densely sintered struts. The paper may be considered for publication in *Nature Communications*, but requires significant revision.

Response: We thank the referee for the recommendation of the publication in *Nature Communications*. Following the referee's comments and suggestions, we have revised our manuscript with the most seriousness (the revised portions are marked in Red Color in the manuscript) and our detailed responses are listed below.

Questions and comments:

1. The presented cellular ceramic structures (see Fig. 4 or Suppl. Fig. 8) could also be produced in a simpler and faster way using direct ink writing (DIW). There is a plethora of literature about that, and e.g. the groups of Jennifer Lewis (Harvard University), Andre Studart (ETH Zürich) or Willenbacher (KIT) have published significant work in that title and the authors should benchmark their new approach against well-established DIW in terms of fabrication speed or accessible feature size. (comment 1) Furthermore, the first paragraph of the Results section should be moved to the Introduction. (comment 2)

Response: We thank the referee very much for the constructive suggestions. **For comment 1:** Yes, we fully agree with you. Ceramic additive manufacturing using direct ink writing (DIW) has been extensively studied, and many research groups have significant contributions

including the groups of Jennifer Lewis, Andre Studart, and Willenbacher. We fully respect these inspiring studies. In the revised manuscript, we have referred to these published significant works about ceramic additive manufacturing using DIW, and compared our new STATS approach with the well-established DIW, SLA/DLP, and SLS in terms of effective constituent loading, fabrication speed, and accessible feature size, as shown in Supplementary Table 1¹⁻¹². All these references have been cited in the revised manuscript.

The slurry extrusion-based DIW has the advantage of high effective constituent loading regardless of its limited fabrication speed and feature size. As a comparison, the photosensitive resin-based SLA/DLP possesses an increasing fabrication speed and decreasing feature size, with limited effective constituent loading. The proposed STATS approach combines the advantages of DIW in effective constituent loading and SLA/DLP in fabrication speed and accessible feature size. Our approach utilizes SLA/DLP to process high-resolution organic lattices, and we prepare the feedstock suspensions with highly effective ceramic constituent loading, which is usually used in DIW. Moreover, the exposure time of common resin is usually 2-3 s per layer in DLP, while it increases over one time (5-7 s) for the curing of DLP processed ceramic-resin mixture. In the proposed STATS approach, the organic lattices with designed architectures are processed by the additive manufacturing of common resin, greatly increasing the printing speed. Therefore, the separation of ingredient synthesis from architecture building in the STATS manufacturing process dramatically simplifies and speeds up the manufacturing process of cellular ceramics.

Supplementary Table 1. Comparison of the proposed STATS process with the DIW, SLA, DLP, and SLS.

3D printing method	References	Materials	Effective constituent loading	Fabrication speed	Accessible feature size
DIW	Dutto et al. ¹	Clay	20-50 wt%	5.6-62.9 mm s ⁻¹	1.6-6.3 mm lateral and 2-4.5 mm height
	Maurath et al. ²	Al ₂ O ₃	23-31 vol%	42-55 mm s ⁻¹	0.2-0.62 mm lateral and 2-4.5 mm height
	Smay et al. ³	PZT	47 vol%	6 mm s ⁻¹	0.2-0.4 mm lateral and 0.16-0.32 mm height

	Menne et al. ⁴	BaTiO ₃	31 vol%	10 mm s ⁻¹	0.07-0.15 mm lateral and 0.12 mm height
SLA	Griffith et al. ⁵	Al ₂ O ₃	40-50 vol%	NA	0.2-0.4 mm height
	Liu et al. ⁶	ZrO ₂ -Al ₂ O ₃	46.8 vol%	8 m s ⁻¹	0.04 mm lateral
	Chen et al. ⁷	BaTiO ₃	70 wt%	NA	0.02 mm lateral and 0.05 mm height
DLP	Chen et al. ⁸	BaTiO ₃	40 vol%	NA	0.01 mm height
	Komissarenko et al. ⁹	ZrO ₂	33 vol%	5 s per layer	0.05 mm height
	Varghese et al. ¹⁰	Al ₂ O ₃	45-55 wt%	5-7 s per layer	0.045 mm lateral and 0.025-0.05 mm height
SLS	Bertrand et al. ¹¹	ZrO ₂ -Y ₂ O ₃	NA	1.25-2 mm s ⁻¹	0.52-1 mm lateral
	Zhang et al. ¹²	BaTiO ₃	NA	0.5 mm s ⁻¹	1 mm lateral
STATS process	This work	PZT	> 90 wt% for dried gel	2 s per layer for organic lattice	0.065-0.2 mm shell thickness

The following sentences are added to the revised manuscript (page 4):

“Supplementary Table 1 summarizes the parameters in terms of effective constituent loading, fabrication speed, and accessible feature size for the well-established DIW, SLA/DLP, and SLS³¹⁻⁴². The slurry extrusion-based DIW has the advantage of high effective constituent loading regardless of its limited fabrication speed and feature size, while the photosensitive resin-based SLA/DLP possesses an increasing fabrication speed and decreasing feature size, with limited effective constituent loading.

In comparison, the philosophy of the STATS process is different. It avoids an engineering trade-off by separating the steps of architecture building and ingredient synthesis. The proposed STATS approach combines the advantages of DIW in effective constituent loading and SLA/DLP in fabrication speed and accessible feature size. We utilize SLA/DLP to process high-resolution organic lattices, and we prepare the feedstock suspensions, similar to the DIW process, with high effective ceramic constituent loading. Moreover, the exposure time of common resin is usually 2-3 s per layer in DLP, while it increases over one time (5-7 s) for the

curing of DLP processed ceramic-resin mixture⁴⁰. In the proposed STATS approach, the organic lattices with designed architectures are processed by the additive manufacturing of common resin, greatly increasing the printing speed. Therefore, the separation of ingredient synthesis from architecture building in the STATS manufacturing process dramatically simplifies and accelerates the cellular ceramic manufacturing.”

For comment 2: The first paragraph of the Results section has been moved to the Introduction section. Thank you for your suggestion.

2. The wording in the section about the design principle on page 5 could be clearer e.g. in line 211 it should read “...increasing precursor solution concentration...” (comment 1) and the term “capillary length” is not clearly defined (line 224) (comment 2).

Response: We thank the referee for this comment. **For comment 1:** The following sentence is revised in the manuscript (page 6):

“The contact angle increases, accompanied by the increasing concentration of the precursor solution, but all within 90° (Fig. 3b and Supplementary Fig. 13).”

For comment 2: Capillary length is a characteristic length scale for fluid subject to a body force from gravity and a surface force due to surface tension¹³, which is most commonly given by

$$\lambda = \sqrt{\sigma / \Delta \rho g} \quad (\text{R1})$$

where λ is the surface tension of the fluid interface, g is the gravitational acceleration and $\Delta \rho$ is the mass density difference of the fluids.

The following sentence is revised in the manuscript (page 6):

“Besides, the size of the created fluid interface ($a - 2r$) must be within the capillary length $\sqrt{\sigma_{12} / \Delta \rho_{12} g}$, where σ_{12} is the interfacial tension of fluid 1-fluid 2, $\Delta \rho_{12}$ is the density difference between fluid 1 and fluid 2, and g is the gravitational acceleration²⁵.”

The following sentence is revised in Supporting Information (page 6):

“In addition, the size of the created fluid interface must be within the capillary length, a characteristic length scale for the fluid subject to a body force from gravity and a surface force due to surface tension²⁵, which means

$$a-2r < \sqrt{\sigma_{12}/\Delta\rho_{12}g} \quad (9)$$

where $\Delta\rho_{12}$ is the density difference between fluid 1 and fluid 2, and g is the gravitational acceleration.”

3. The authors have varied the precursor solution concentration in order to change the wetting angle. This may have an effect on drying time, gel-structure, sintering conditions and also the properties of the final ceramics. (comment 1)

Alternatively, the wetting angle could be changed in a wide range by adding trace amounts of appropriate surfactants to a precursor solution with fixed concentration, also the surface properties of the 3D printed organic scaffold may be varied (e.g. plasma treatment) to tune the wetting behavior, these aspects should be discussed carefully. (comment 2)

Response: We thank the referee very much for the constructive suggestions. **For comment 1:** Yes, we fully agree with you. The precursor solution concentration does have an effect on drying time, gel-structure, sintering conditions and also the properties of the final ceramics. For instance, the reduced precursor solution concentration leads to an increasing drying time during the sol-gel transition, as well as a reduced shell thickness for the sintered cellular ceramics, which decreases from 187 μm to 100 μm as the concentration decreases from 5 mM to 1 mM (Fig. 2e and Supplementary Fig. 7).

The following sentence is added in Supporting Information (page 5):

“The shell thickness is significantly influenced by the concentration of the precursor solution, which increases from 100 μm to 187 μm as the concentration increases from 1 mM to 5 mM (Fig. 2e and Supplementary Fig. 7).”

For comment 2: The addition of surfactants in the precursor solution and the plasma treatment on the organic lattices also benefit the decrease of contact angle to some extent. For instance,

under the condition of 2 mM concentration of the precursor solution, the contact angle is 20° while the addition of 1 wt.% surfactants in the precursor solution reduces the contact angle to 13° and a plasma treatment (2 min) on the organic lattices reduces it to 15°. Considering that a contact angle of 20° is enough to form a solution-philic condition (α is 0.98 at the contact angle of 20°, close to its maximum), these factors are not considered in the following experiments from the perspective of simplification of experimental procedure.

The following sentence is added in the manuscript (page 6):

“Besides, the addition of surfactants in the precursor solution and the plasma treatment on the organic lattices also benefit the decrease of contact angle to some extent (Supplementary Fig. 14). Considering that a contact angle of 20° is enough to form a solution-philic condition, these factors are not considered in the following experiments from the perspective of simplification of experimental procedures.”

The following sentence is added in Supporting Information (page 2):

“The addition of surfactants in the precursor solution and the plasma treatment on the organic lattices also benefit the decrease of contact angle to some extent. Under the condition of 2 mM concentration of the precursor solution, the contact angle is 20° while the addition of 1 wt.% surfactants in the precursor solution reduces the contact angle to 13° and a plasma treatment (2 min) on the organic lattices reduces it to 15°. Considering that a contact angle of 20° is enough to form a solution-philic condition (α is 0.98 at the contact angle of 20°, close to its maximum), these factors are not considered in the following experiments from the perspective of simplification of experimental procedures.”

Fig. 2e. The variation of shell thickness of the sintered cellular ceramics along with the precursor solution concentration, at a 1 mm frame length and 0.1 mm frame radius of the organic lattice. Insets show the SEM images of the cellular ceramics with shell structure and layer-by-layer characteristic. Scale bar, 200 μm .

Supplementary Fig. 7. SEM images of the sintered PZT cellular ceramics prepared from the precursor solution with various concentrations from **a**, 5 mM; **b**, 4 mM; **c**, 3mM; **d**, 2 mM; **e**, 1 mM. The organic lattice has a 1 mm frame length and a 100 μm frame radius.

Supplementary Fig. 14. Contact angle of the precursor solution on the resin plate in air. **a**, Contact angle (15°) after a plasma treatment (2 min) on the resin plate. **b**, Contact angle (13°) after the addition of 1 wt.% surfactants in the precursor solution.

4. Caption of Fig. 3 is incomplete: The insert in Fig. 3d is not visible (not even when enlarged on the computer screen) (comment 1), the meaning of the different lines in Fig. 3d is unclear (comment 2). The colors of the lines in Fig. 3e are not explained, correlation to frame radius? (comment 3) Meaning of blue regions in Fig. 3h is not explained, meaning of subfigures i), ii), iii, and iv) in Fig. 3h is not explained (comment 4).

Response: We thank the referee very much for the careful check and valuable comments. **For comment 1:** The insert in Fig. 3d is revised as following:

Fig. 3d, Theoretical and experimental results showing the influence of frame length a and radius r on the liquid capture. The blue and yellow regions respectively include the appropriate geometry parameters for successful liquid capture in unit cell and liquid arrangement in architected lattice from theoretical analysis. The red circle and blue cross icons indicate successful and failed liquid capture in the experiments, respectively. Insets show the optical image of the experimental results (black dash line) with various frame lengths from 0.4 mm to 2.2 mm and the same radius of 0.1 mm.

For comment 2: The different lines in Fig. 3d include black solid line ($a/r = 2$), blue solid line ($a/r = 14.3$), red solid line ($a - 2r = 1.64$ mm), blue dash line ($a/r = 7.15$), red dash line ($a - 2r = 0.82$ mm).

Specifically, combining Equation 1 and the geometry requirement ($a > 2r$), we obtain $2.0 < a/r < 14.3$, shown as Area I between the **blue solid line** and **black solid line** in Figure 3d. The

obtained result ($a - 2r < 1.64$ mm) is shown as Area II below the **red solid line** in Figure 3d, and the intersection between Area I and Area II (blue region) represents the safe area with the appropriate frame length and radius to successfully create a stable liquid interface.

When considering the whole architected lattice, in which the precursor solution is assembled with a programmed arrangement, both of the successful capture in parts of the unit cells and the failed capture in other parts are required. Taking the architected lattices depicted in Fig. 3g for example, the unit cells with a length of a should be filled with the precursor solution while it is the opposite for the cells with a length of $i) 2a$. As a result, the constrains, shown as the yellow region in Fig. 3d, consisting of $7.15 < a/r < 14.3$ (**blue dash line** and blue solid line) and $0.82 \text{ mm} < a - 2r < 1.64 \text{ mm}$ (**red dash line** and red solid line), are required to form a programmed arrangement with an interval length of $i) 2a$ depicted in Fig. 3g.

The following sentences are revised in the manuscript (page 6):

“Combining Equation 1 and the geometry requirement ($a > 2r$), we obtain $2.0 < a/r < 14.3$, shown as Area I between the blue solid line and black solid line in Fig. 3d.”

“The obtained result ($a - 2r < 1.64$ mm) is shown as Area II below the red solid line in Figure 3d, and the intersection between Area I and Area II (blue region) represents the safe area with the appropriate frame length and radius to successfully create a stable liquid interface.”

The following sentence is revised in Supporting Information (page 7):

“As a result, the constrains, shown as the yellow region in Fig. 3d, consisting of $7.15 < a/r < 14.3$ (blue dash line and blue red solid line) and $0.82 \text{ mm} < a - 2r < 1.64 \text{ mm}$ (red dash line and red solid line), are required to form a programmed arrangement with an interval length of $i) 2a$ depicted in Fig. 3g.”

For comment 3: The colors of the lines in Fig. 3e correlate with frame radius, namely, the red, blue, green and purple line corresponds to a radius of 0.1 mm, 0.15 mm, 0.2 mm, 0.25 mm, respectively.

Fig. 3e summarizes the relation between liquid height and the maximum contact perimeter in a unit column as the radius increases from 0.1 mm (red dash line) to 0.25 mm (purple dash line)

(detailed in Supplementary Fig. 17c). A relation between the liquid height h and the contact perimeter s with corresponding radii from the gravitational force is also given as the solid line in Fig. 3e (e.g., red solid line corresponds to a radius of 0.1 mm) according to Equation 2.

The following sentences are revised in the manuscript (page 7):

“A relation between the liquid height h and the contact perimeter s with corresponding radii from the gravitational force is also given as the solid line in Fig. 3e (e.g., red solid line corresponds to a radius of 0.1 mm) according to Equation 2.”

For comment 4: The blue region in Fig. 3h represents the fraction of precursor solution in the FE analysis, in which a darker color means a higher proportion. And the subfigures i), ii), iii), and iv) in Fig. 3h represent the formation of the programmed liquid arrangement from i) to iv) under gravity for the architected lattice with an interval length of $3a$.

The following sentence is revised in the manuscript (page 14):

“Finite element analysis of the formation of the programmed liquid arrangement from i) to iv) under gravity for the architected lattice with an interval length of $3a$.”

Fig. 3c. FE results of the dependence of α on the contact angle θ_{123} for precursor solution in air. Insets show the liquid distribution at the contact angle of 20° and 60° .

Fig. 3h, Finite element analysis of the formation of the programmed liquid arrangement from i) to iv) under gravity for the architected lattice with an interval length of $3a$.

5. The section about the piezoelectric performance of the cellular ceramic metamaterials is the weakest part of the manuscript. It is not clear what could be achieved here beyond the state of the art regarding piezoelectric properties, e.g. in David Menne et al., ACS Applied Materials and Interfaces 14 (2), 3027-3037 (2022) cellular piezoceramic structures are described with higher d_{33} values (comment 1) and it is demonstrated in Fig. S1 of that paper, that strut diameters as small as $70\ \mu\text{m}$ can be achieved using DIW. (comment 2)
6. Fig. 4 is confusing, in Fig. 4a a d_{33} value of about $500\ \text{pC/N}$ is claimed (red bar) but in Fig. 4b d_{33} values are only about $200\ \text{pC/N}$ which is clearly lower than the value of $350\ \text{pC/N}$ reported by David Menne et al. (see above) (comment 1) On page 7, line 287 a pressure sensitivity of $3.2\ \text{mV/kPa}$ is mentioned (slope of Fig 4c) the authors should explain how this quantity is related to the extensively discussed g_{33} data. (comment 3)

Response: We thank the referee very much for these constructive comments. **For comment 1:** The value of effective d_{33} significantly depends on the porosity of the cellular piezoceramics. In our work, we prepare two types of PZT piezoceramics through the STATS manufacturing process, with solid and cellular structures, respectively. The as-prepared solid piezoceramic plates exhibit a piezoelectric constant (d_{33}) around $500\ \text{pC N}^{-1}$, close to commercial PZT (Fig. 4a). To further demonstrate the advantages of the STATS approach in manufacturing cellular piezoceramics, we fabricate a series of cellular PZT ceramics with various relative densities (relative to the density of solid ceramics) and characterize their piezoelectric properties. The relative density of 0.13, 0.21, 0.27, and 0.34 corresponds to a porosity of 0.980, 0.963, 0.951 and 0.931, respectively (Supplementary Fig. 21a). Owing to the similar configuration and porosity, the cellular PZT ceramics exhibit a close d_{33} ($\sim 200\ \text{pC N}^{-1}$) regardless of the changing relative density (Supplementary Fig. 21b). Compared with the solid piezoceramic plates, the decreased d_{33} is attributed to the significantly increased porosity.

Generally speaking, a lower porosity leads to a higher d_{33} . In the work reported by David Menne et al., the highest d_{33} ($330\ \text{pC N}^{-1}$) is achieved at a porosity of 0.58. This porosity is

obviously lower than that in our work (> 0.9), and the d_{33} is larger than that in our work (~ 200 pC N $^{-1}$). However, as the porosity increases to 0.63, the d_{33} value dramatically reduces to 101 pC N $^{-1}$, obviously lower than that in our work. This occurs due to the significantly increased strut porosity. As a comparison, the locally compact structure of the cellular piezoceramics from the STATS process in our work ensures an extremely low strut porosity, finally resulting a high d_{33} (~ 200 pC N $^{-1}$) even at a relatively high porosity (> 0.9).

The advantage of the STATS process proposed in our work lies in the manufacturing of globally porous but locally compact structure, endowing the cellular piezoceramics with a relatively low density and a higher piezoelectric constant than a variety of piezoelectric ceramics/composites. As a result, the locally compact structure in our work ensures a dense strut with low strut porosity, which allows the cellular piezoceramics to possess a relatively high d_{33} (~ 200 pC N $^{-1}$) even at a very high overall porosity (> 0.9).

The following sentences are revised in the manuscript (page 14):

“The piezoelectric performance of the prepared cellular PZT ceramics is demonstrated by characterizing their d_{33} and piezoelectric voltage coefficient (g_{33}). Owing to the similar configuration and porosity (from 93.1% to 98.0%), the cellular PZT ceramics exhibit a close d_{33} (~ 200 pC N $^{-1}$) regardless of the changing relative density (Supplementary Fig. 21). Compared with the solid piezoceramic plates, the decreased d_{33} is attributed to the high porosity. Nevertheless, the locally compact structure ensures that the STATS manufactured cellular piezoceramics possess a relatively high d_{33} even at a very high overall porosity ($> 90\%$).”

Fig. 4a. Comparison of the piezoelectric charge coefficient d_{33} between the STATS manufactured piezoceramics including solid piezoceramic plates and cellular piezoceramics with high porosity and the previously reported architected piezoelectric composites and ceramics.

Supplementary Fig. 21. a, The corresponding porosity to the relative density, in which the relative density of 0.13, 0.21, 0.27, and 0.34 corresponds to a porosity of 0.980, 0.963, 0.951 and 0.931, respectively. **b,** The effective d_{33} and relative permittivity of the prepared cellular piezoceramics with a relative density of 0.13, 0.21, 0.27 and 0.34.

For comment 2: In the work reported by David Menne et al., they utilize DIW to print with a strut/pore diameter of approx. 1:6 (70 μm strut and 440 μm pore diameter). In our work, the shell thickness and chamber length depend on the feature size of organic lattice and the precursor solution concentration. The shell thickness is significantly influenced by the concentration of the precursor solution, which increases from 100 μm to 187 μm as the concentration increases from 1 mM to 5 mM (Fig. 2e and Supplementary Fig. 7). Besides the concentration of the precursor solution, the shell thickness and chamber length of the sintered cellular ceramics also depend on the feature size of organic lattice (*i.e.*, frame length and radius). For a lattice assembled by cube cell (a frame length of 1 mm and a frame radius of 0.1 mm) and a 1 mM precursor solution concentration, the sintered ceramics possess a shell thickness of 0.1 mm and chamber length of 0.76 mm, with a length/thickness ratio of 7.6. As the frame length increases to 1.25 mm and 1.5 mm with a frame radius of 100 μm , a chamber length of 0.83 mm and 0.99 mm with a shell thickness of 90 μm are obtained in the sintered cellular ceramics, achieving a larger length/thickness ratio of 9.2 and 11, respectively (Supplementary Fig. 8a&b). As the frame radius decreases to 75 μm with a frame length of 1

mm, a chamber length of 0.65 mm and a shell thickness of 65 μm are obtained in the sintered cellular ceramics, with a larger length/thickness ratio of 10 (Supplementary Fig. 8c).

Moreover, in the work reported by David Menne et al., the printed strut possesses a high porosity from 0.3 to 0.5, which adversely affects the mechanical and piezoelectric properties of the printed cellular piezoceramics. As the overall porosity increases to 0.63, the d_{33} value dramatically reduces to 101 pC N^{-1} , lower than 200 pC N^{-1} even at a higher overall porosity (> 90%) in our work. The locally compact structure in our work ensures a dense strut with low strut porosity, which allows the cellular piezoceramics to possess a relatively high d_{33} at a very high overall porosity.

The following sentences are revised in the manuscript (page 5):

“The shell thickness is significantly influenced by the concentration of the precursor solution, which increases from 100 μm to 187 μm as the concentration increases from 1 mM to 5 mM (Fig. 2e and Supplementary Fig. 7). Besides the concentration of the precursor solution, the shell thickness and chamber length of the sintered cellular ceramics also depend on the feature size of organic lattice (*i.e.*, frame length and radius). Once the frame radius of the organic lattice is reduced from 100 μm to 75 μm , a minimum shell thickness of 65 μm is achieved with a length/thickness ratio of 10 (Supplementary Fig. 8).”

“Nevertheless, the locally compact structure ensures that the STATS manufactured cellular piezoceramics possess a relatively high d_{33} even at a very high overall porosity (> 90%).”

Supplementary Fig. 8. **a**, SEM image of the sintered cellular ceramics with a chamber length of 0.99 mm and a shell thickness of 90 μm , at a frame length of 1.5 mm and a frame radius of 100 μm of the organic lattice. **b**, SEM image of the sintered cellular ceramics with a chamber length of 0.83 mm and a shell thickness of 90 μm , at a frame length of 1.25 mm and a frame radius of 100 μm of the organic

lattice. **c**, SEM image of the sintered cellular ceramics with a chamber length of 0.65 mm and a shell thickness of 65 μm , at a frame length of 1 mm and a frame radius of 75 μm of the organic lattice. The concentration of precursor solution is 1 mM. Scale bar, 200 μm .

For comment 3: The electrical responses of the piezoelectric materials under mechanical deformation originate from the direct piezoelectric effect. Under compressive pressure, the d_{33} constants of the cellular PZT ceramics are quantified by the ratio of the applied load F_N and the generated charge Q , which is given by $d_{33}=Q/F_N$. And the g_{33} constants are further quantified by the ratio of electric permittivity ϵ_{33} and d_{33} constant ($g_{33}=d_{33}/\epsilon_{33}$). The effective permittivity ϵ_{33} can be calculated by $\epsilon_{33}=Cl/A$, where C is the capacitance of the cellular PZT ceramics, l is the distance between the electrodes and A is the cross-section area of the cellular PZT ceramics. In addition, the generated charge Q can be expressed by $Q=CV_{out}$, and the applied load F_N can be expressed by $F_N=PA$, where V_{out} is the open-circuit output voltage and P is the applied pressure. Combining these equations, the open-circuit output voltage can be given by

$$\frac{V_{out}}{P}=g_{33} l \quad (\text{R1})$$

As a result, the open-circuit output voltage is linearly correlated with the applied pressure, and a larger g_{33} constant leads to an increasing output voltage.

The calculated open-circuit output voltage according to Equation R1 should be under quasi-static state, usually larger than the output voltage measured in practical application. Nevertheless, this equation is helpful to understand that a larger g_{33} constant leads to an increasing pressure sensitivity dV_{out}/dP .

The following sentences are added in Supporting Information (pages 8-9):

“The electrical responses of the piezoelectric materials under mechanical deformation originate from the direct piezoelectric effect. Under compressive pressure, the d_{33} constants of the cellular PZT ceramics are quantified by the ratio of the applied load F_N and the generated charge Q , which is given by $d_{33}=Q/F_N$. And the g_{33} constants are further quantified by $g_{33}=d_{33}/\epsilon_{33}$. The effective permittivity ϵ_{33} can be calculated by $\epsilon_{33}=Cl/A$, where C is the

capacitance of the cellular PZT ceramics, l is the distance between the electrodes and A is the cross-section area of the cellular PZT ceramics. In addition, the generated charge Q can be expressed by $Q=CV_{out}$, and the applied load F_N can be expressed by $F_N=PA$, where V_{out} is the open-circuit output voltage and P is the applied pressure. Combining these equations, the open-circuit output voltage can be given by

$$\frac{V_{out}}{P}=g_{33} l \quad (15)$$

As a result, the open-circuit output voltage is linearly correlated with the applied pressure, and a larger g_{33} constant leads to an increasing output voltage.

The calculated open-circuit output voltage according to Equation 15 should be under quasi-static state, usually larger than the output voltage measured in practical application. Nevertheless, this equation is helpful to understand that a larger g_{33} constant leads to an increasing pressure sensitivity dV_{out}/dP .”

References

1. Dutto A, Zanini M, Jeoffroy E, et al. 3D Printing of Hierarchical Porous Ceramics for Thermal Insulation and Evaporative Cooling[J]. *Advanced Materials Technologies*, 2023, 8(4): 2201109.
2. Maurath J, Willenbacher N. 3D printing of open-porous cellular ceramics with high specific strength[J]. *Journal of the European Ceramic Society*, 2017, 37(15): 4833-4842.
3. Smay J E, Cesarano J, Lewis J A. Colloidal inks for directed assembly of 3-D periodic structures[J]. *Langmuir*, 2002, 18(14): 5429-5437.
4. Menne D, Lemos da Silva L, Rotan M, et al. Giant functional properties in porous electroceramics through additive manufacturing of capillary suspensions[J]. *ACS Applied Materials & Interfaces*, 2022, 14(2): 3027-3037.
5. Griffith M L, Halloran J W. Freeform fabrication of ceramics via stereolithography[J]. *Journal of the American Ceramic Society*, 1996, 79(10): 2601-2608.
6. Liu X, Zou B, Xing H, et al. The preparation of ZrO₂-Al₂O₃ composite ceramic by SLA-3D printing and sintering processing[J]. *Ceramics International*, 2020, 46(1): 937-944.
7. Chen Z, Song X, Lei L, et al. 3D printing of piezoelectric element for energy focusing and ultrasonic sensing[J]. *Nano Energy*, 2016, 27: 78-86.
8. Chen X, Sun J, Guo B, et al. Effect of the particle size on the performance of BaTiO₃ piezoelectric ceramics produced by additive manufacturing[J]. *Ceramics International*, 2022, 48(1): 1285-1292.

9. Komissarenko D A, Sokolov P S, Evstigneeva A D, et al. DLP 3D printing of scandia-stabilized zirconia ceramics[J]. *Journal of the European Ceramic Society*, 2021, 41(1): 684-690.
10. Varghese G, Moral M, Castro-García M, et al. Fabrication and characterisation of ceramics via low-cost DLP 3D printing[J]. *Boletín de la Sociedad Española de Cerámica y Vidrio*, 2018, 57(1): 9-18.
11. Bertrand P, Bayle F, Combe C, et al. Ceramic components manufacturing by selective laser sintering[J]. *Applied Surface Science*, 2007, 254(4): 989-992.
12. Zhang X, Wang F, Wu Z, et al. Direct selective laser sintering of hexagonal barium titanate ceramics[J]. *Journal of the American Ceramic Society*, 2021, 104(3): 1271-1280.
13. Zhang, Y. *et al.* Magnetic-actuated ‘capillary container’ for versatile three-dimensional fluid interface manipulation. *Sci Adv* 7, (2021).

Reviewer #2 (Remarks to the Author):

The manuscript discusses a novel approach for manufacturing cellular ceramics inspired by the biomineralization process found in nature. This approach involves separating ingredient synthesis from architecture building to create programmable 3D configurations of cellular ceramics. The technique utilizes surface tension to capture precursor solutions in architected cellular lattices, allowing for precise control of liquid geometry. The authors investigate the geometry parameters of these lattices theoretically and experimentally to guide 3D fluid interface creation. They successfully manufacture various cellular ceramics, including piezoceramics with enhanced properties, such as a higher piezoelectric constant and designed anisotropy.

The proposed approach possesses similarities to the negative replica approach used for foams as well as for printed sacrificial structures (see for example: <https://doi.org/10.1111/j.1151-2916.1997.tb02993.x>); nevertheless, the theoretical approach proposed here allows to predict and design structures specifically to fit the process requirements and is certainly novel.

Response: We thank the referee very much for the recognition of the novelty of the proposed approach in this work. For the referee's concern about the similarities to the negative replica approach (the mentioned reference has been cited in the manuscript), the following elaboration may respond to this concern. In traditional negative replica approach for porous ceramic manufacturing¹, the remove of organic binder in feedstock slurry during sintering leads to the micropores inside the ceramic struts, greatly influence the quality and performance of the produced ceramics. As a comparison, the proposed surface-tension-driven method in our work utilizes surface tension to capture the precursor solution. The volume shrink and structure reconfiguration (from droplet to solid shell) during sol-gel transition lead to a generally porous but locally compact ceramic, contributing to the elimination of micropores.

We also thank the referee for careful reading of our manuscript and providing valuable comments, which helped us to improve the quality of our manuscript. Following the referee's comments and suggestions, we have revised our manuscript with the most seriousness (the revised portions are marked in **Red Color** in the manuscript). Please find our detailed responses below.

Questions and comments:

1. I would recommend to further characterize the cellular structures produce to understand how they fit in the current state of the art; in particular:

- microCT scan and defect assessment; (Comment 1)

- mechanical properties (not only piezoelectric ones) (Comment 2)

Response: We thank the referee very much for the insightful suggestions. **For comment 1:** To further characterize the cellular structure of the STATS manufactured ceramics, X-ray computed microtomography (micro-CT) is conducted. As shown in Supplementary Fig. 9, the micro-CT images clearly show the globally cellular shell structure of the sintered ceramics with a high quality. There are mainly two types of defects, including the cracks existing in the interface between the ceramic shell and the removed organic lattices, and the extra ceramic walls between the unit cells. These defects have little influence on the overall performance of the sintered cellular ceramics.

The following sentences are added in the manuscript (page 5):

“To further characterize the cellular structure of the STATS manufactured ceramics, X-ray computed microtomography (micro-CT) is conducted. As shown in Supplementary Fig. 9, the micro-CT images clearly show the globally cellular shell structure of the sintered ceramics with a high quality. There are mainly two types of defects, including the cracks existing in the interface between the ceramic shell and the removed organic lattices, and the extra ceramic walls between the unit cells. These defects have little influence on the overall performance of the sintered cellular ceramics.”

Supplementary Fig. 9. X-ray computed microtomographic (micro-CT) images of the sintered cellular PZT ceramics. **a**, Reconstructed model of the ceramic sample. **b&c**, Cross-section view from **b**, slice 1 and **c**, slice 2. Scale bar, 1 mm. **d**, **e**, & **f**, Single slices from **d**, slice 1 and **e**, slice 2 and **f**, slice 3. The micro-CT images clearly show the globally cellular shell structure of the sintered ceramics with a high quality. There are mainly two types of defects, including the cracks existing in the interface between the ceramic shell and the removed organic lattices, and the extra ceramic walls between the unit cells. Scale bar, 1 mm.

For comment 2: To further demonstrate the advantages of the STATS approach in manufacturing cellular piezoceramics, we fabricate a series of cellular PZT ceramics with various relative densities (relative to the density of solid ceramics) and characterize their mechanical properties. Under a uniaxial compression test, the fracture strength of the cellular PZT ceramics increases from 1.0 MPa to 3.9 MPa as the relative density increases from 0.13 to 0.34 (Fig. 4b and Supplementary Fig. 19). For type ii and type iii cellular PZT ceramics with varied configurations along the three principal directions, there is an obvious difference for the fracture strength under pressure from different directions, demonstrating the mechanical anisotropy (Supplementary Fig. 22). We further characterize the fracture strength of various cellular ceramic (e.g., 1.6 MPa for BaTiO₃, 5.8 MPa for Al₂O₃), comparable to the porous ceramics previously reported (Supplementary Fig. 20)².

The following sentences are revised in the manuscript (page 3):

“Under uniaxial compression test, the fracture strength of the cellular PZT ceramics increases from 1.0 MPa to 3.9 MPa as the relative density increases from 0.13 to 0.34 (Fig. 4b and Supplementary Fig. 19). The fracture strength of the cellular ceramics with different constituents is further characterized (*e.g.*, 1.6 MPa for BaTiO₃ and 5.8 MPa for Al₂O₃, as shown in Supplementary Fig. 20), comparable to the 3D printed porous ceramics previously reported.”

“However, for type ii and type iii cellular ceramics with varied configurations along the three principal directions, besides mechanical anisotropy (Supplementary Fig. 22), there is an obvious difference for the output voltages under pressure from different directions, demonstrating the piezoelectric anisotropy.”

Fig. 4b. The effective g_{33} constant and fracture strength of the prepared cellular piezoceramics with a relative density of 0.13, 0.21, 0.27 and 0.34.

Supplementary Fig. 19. The compressive stress-strain curves of the prepared cellular PZT ceramics with a relative density of **a**, 0.13; **b**, 0.21; **c**, 0.27; and **d**, 0.34.

Supplementary Fig. 20. a, The relative densities and fracture strength of the prepared Al₂O₃ and BaTiO₃ cellular ceramics. **b**, The compressive stress-strain curves of the prepared Al₂O₃ and BaTiO₃ cellular piezoceramics.

Supplementary Fig. 22. The compressive stress-strain curves of the prepared cellular PZT ceramics with **a**, Type ii configuration and **b**, Type iii configuration.

2. I would also better address the role of the usage of sol-gel precursors in achieving the proposed results: what happens if conventional powder-based suspensions are used instead?

Response: We thank the referee for the useful suggestion. As a comparison, we prepared the cellular ceramics based on the conventional powder-binder-water based suspensions. Usually, the conventional powder-binder-water based suspensions are developed by combining the fine ceramic powder particles with the binder materials in deionized water in the presence of suitable additives (e.g., surfactant).

Here, to prepare the conventional powder-binder-water based suspensions, the PZT powders are dispersed in deionized water to form a suspension with solid loading level of 40 vol%, with polyvinyl alcohol (PVA) and polyacrylic acid which serve as the binder and dispersant respectively. The prepared suspension is ball-milled for 24 hours in zirconia media to create a homogeneous and fine suspension, which is used in the following surface-tension-assisted two-step (STATS) manufacturing process. Compared to the cellular ceramics from sol-powder suspensions, the sample prepared from the conventional powder-water based suspensions shows a lower quality with more defects (Supplementary Fig. 5).

The following sentences is added in the manuscript (page 5):

“As a comparison, the sample prepared from conventional powder-binder-water based suspensions shows a lower quality with more defects (Supplementary Fig. 5).”

The following sentences are added in Supporting Information (page 3):

“To prepare the conventional powder-binder-water based suspensions, the PZT powders are dispersed in deionized water to form a suspension with solid loading level of 40 vol%, with polyvinyl alcohol (PVA) and polyacrylic acid which serve as the binder and dispersant respectively. The prepared suspension is ball-milled for 24 hours in zirconia media to create a homogeneous and fine suspension, which is used in the following STATS manufacturing process.”

Supplementary Fig.5. SEM images of the PZT cellular ceramics prepared from the conventional powder-binder-water suspensions, showing a lower quality with more defects.

References

1. Bandyopadhyay, Amit , et al. "Processing of Piezocomposites by Fused Deposition Technique." *Journal of the American Ceramic Society* (2005).
2. Mei, H. *et al.* Structure design influencing the mechanical performance of 3D printing porous ceramics. *Ceram Int* **47**, 8389–8397 (2021).

REVIEWERS' COMMENTS

Reviewer #1 (Remarks to the Author):

The authors have carefully and thoroughly addressed all my concerns and issues, the manuscript has gained in quality and clarity.

From my point of view, nothing speaks against publication, except may be for some typos here and there.

Reviewer #2 (Remarks to the Author):

All my comments have been addressed. I have no other comments.

RESPONSE TO REVIEWERS' COMMENTS

Reviewer Comments (and change made in accordance)

Reviewer #1 (Remarks to the Author):

The authors have carefully and thoroughly addressed all my concerns and issues, the manuscript has gained in quality and clarity.

From my point of view, nothing speaks against publication, except may be for some typos here and there.

Response: We thank the referee for recommendation of the publication in *Nature Communications*. Following the referee's comments and suggestions, we have carefully checked the typos through the manuscript and revised them accordingly.

Reviewer #2 (Remarks to the Author):

All my comments have been addressed. I have no other comments.

Response: We thank the referee very much for recommendation of the publication in *Nature Communications*.